# Learning to Decompose and Disentangle Representations for Video Prediction

**Jun-Ting Hsieh**
Stanford University
junting@stanford.edu

**Bingbin Liu**
Stanford University
bingbin@stanford.edu

**De-An Huang**
Stanford University
dahuang@cs.stanford.edu

**Li Fei-Fei**
Stanford University
feifeili@cs.stanford.edu

**Juan Carlos Niebles**
Stanford University
jniebles@cs.stanford.edu

## Abstract

Our goal is to predict future video frames given a sequence of input frames. Despite large amounts of video data, this remains a challenging task because of the high-dimensionality of video frames. We address this challenge by proposing the Decompositional Disentangled Predictive Auto-Encoder (DDPAE), a framework that combines structured probabilistic models and deep networks to automatically (i) decompose the high-dimensional video that we aim to predict into components, and (ii) disentangle each component to have low-dimensional temporal dynamics that are easier to predict. Crucially, with an appropriately specified generative model of video frames, our DDPAE is able to learn both the latent decomposition and disentanglement without explicit supervision. For the Moving MNIST dataset, we show that DDPAE is able to recover the underlying components (individual digits) and disentanglement (appearance and location) as we intuitively would do. We further demonstrate that DDPAE can be applied to the Bouncing Balls dataset involving complex interactions between multiple objects to predict the video frame directly from the pixels and recover physical states without explicit supervision.

## 1   Introduction

Our goal is to build intelligent systems that are capable of visually predicting and forecasting what will happen in video sequences. Visual prediction is a core problem in computer vision that has been studied in several contexts, including activity prediction and early recognition [20, 30], human pose and trajectory forecasting [1, 18], and future frame prediction [22, 31, 39, 44]. In particular, the ability to visually hallucinate future frames has enabled applications in robotics [8] and healthcare [26]. However, despite the availability of a large amount of video data, visual frame prediction remains a challenging task because of the high-dimensionality of video frames.

Our key insight into this high-dimensional, continuous sequence prediction problem is to decompose it into sub-problems that can be more easily predicted. Consider the example of predicting digit movements of Moving MNIST in Figure 1: the transformation that converts an entire frame containing two digits into the next frame is high-dimensional and non-linear. Directly learning such transformation is challenging. On the other hand, if we decompose and understand this video correctly, the underlying dynamics that we must predict are simply the $x, y$ coordinates of each individual digit, which are low-dimensional and easy to model and predict in this case (constant velocity translation).

The main technical challenge is thus: How do we decompose the high-dimensional video sequence into sub-problems with lower-dimensional temporal dynamics? While the decomposition is seemingly

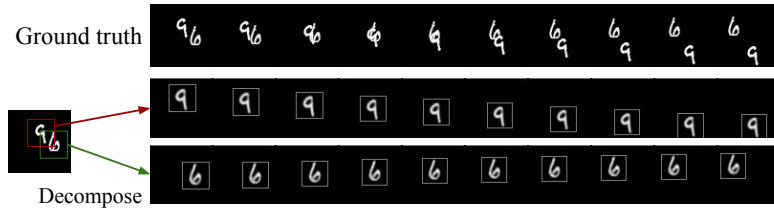

Figure 1: Our key insight is to decompose the video into several components. The prediction of each individual component is easier than directly predicting the whole image sequence. It is important to note that the decomposition is learned automatically without explicit supervision.

obvious in the example from Figure 1, it is unclear how we can extend this to arbitrary videos. More importantly, how do we discover the decomposition *automatically*? It is infeasible or even impossible to hand-craft the decomposition for predicting each type of video. While there have been previous works that similarly aim to reduce the complexity of frame prediction by human pose [38, 42] and patch-based model [22, 31, 40], they either require domain-specific external supervision [38, 42] or do not achieve a significant level of dimension reduction using heuristics [31].

We address this challenge by proposing the Decompositional Disentangled Predictive Auto-Encoder (DDPAE), a framework that combines structured probabilistic models and deep networks to automatically (i) decompose the video we aim to predict into components, and (ii) disentangle each component into low-dimensional temporal dynamics that are easy to predict. With appropriately specified generative model on future frames, DDPAE is able to learn both the video decomposition and the component disentanglement that are effective for video prediction without any explicit supervision on these latent variables. By training a structural generative model of future frames like DDPAE, the aim is not only to obtain good future frame predictions, but also to learn to produce good decomposition and understanding of videos that significantly reduce the complexity of visual frame prediction.

We evaluate DDPAE on two datasets: Moving MNIST [31] and Bouncing Balls [3]. Moving MNIST has been widely used for evaluating video prediction models [15, 31, 43]. We show that DDPAE is able to learn to decompose videos in the Moving MNIST dataset into individual digits, and further disentangles each component into the digit's appearance and its spatial location which is much easier to predict (Figure 1). This significantly reduces the complexity of frame prediction and leads to strong quantitative and qualitative improvements over the baselines that aim to predict the video as a whole [6, 37]. We further demonstrate that DDPAE can be applied to the Bouncing Balls dataset, which has been used mainly for approaches that have access to full physical states (location, velocity, mass) [2, 3, 9]. We show that DDPAE is able to achieve reliable prediction of such complex systems *directly from pixels*, and recover physical properties without explicitly modeling the physical states.

## 2 Related Work

**Video Prediction.** The task of video prediction has received increasing attention in the community. Early works include prediction on small image patches [28, 31]. Recent common approaches for full frame prediction predict the feature representations that generate future frames [6, 22, 23, 37, 38, 39] in a sequence-to-sequence framework [4, 32], which has been extended to incorporate spatio-temporal recurrence [15, 31, 43]. Instead of directly generating the pixels, transformation-based models focus on predicting the difference/transformation between frames and lead to sharper results [5, 8, 21, 35, 39, 40, 44, 45]. We also aim to predict the transformation, but only for the temporal dynamics of the decomposed and disentangled representation, which is much easier to predict than whole-frame transformation.

**Visual Representation Decomposition.** Decomposing the video that we aim to predict into components plays an important role to the success of our method. The idea of visual representation decomposition has also been applied in different contexts, including representation learning [27], physics modeling [3], and scene understanding [7]. In particular, some previous works use methods such as Expectation Maximization to perform perceptual grouping and discover individual objects in videos [11, 12, 36].

A highly related work is Attend-Infer-Repeat (AIR) by Eslami *et al.* [7], which decomposes images in a variational auto-encoder framework. Our work goes beyond the image and extends to the temporal dimension, where the model automatically learns the decomposition that is best suited for predicting the future frames. Concurrent to our work, Kosiorek *et al.* [19] proposed the Sequential Attend-Infer-Repeat (SQAIR), which extends the AIR model and is very similar to our work.

**Disentangled Representation.** To learn meaningful decomposition, our DDPAE enforces the components to be disentangled into a representation with low-dimensional temporal dynamics. The idea of disentangled representation has already been explored [6, 34, 37] for video. Denton *et al.* [6] proposed DRNet, where representations are disentangled into content and pose, and the poses are penalized for encoding semantic information with the use of a discrimination loss. Similarly, MCNet [37] disentangles motion from content using image differences and shared a single content vector in prediction. Note that some videos are hard to directly disentangle. Our work addresses this by decomposing the video so that each component can actually be disentangled.

**Variational Auto-Encoder (VAE).** Our DDPAE is based on the VAE [17], which provides one solution to the multiple future problem [42, 44]. VAEs have been used for image and video generation [7, 13, 28, 29, 33, 41, 42, 44]. Our key contribution is to make the model *structural*, where the latent representation is decomposed and more importantly disentangled. Our network models both motion and content probabilistically, and is regularized by learning transformations in a way similar to [16].

## 3 Methods

Our goal is to predict $K$ future frames given $T$ input frames. Our core insight is to combine structured probabilistic models and deep networks to (i) decompose the high-dimensional video into components, and (ii) disentangle each component into low-dimensional temporal dynamics that are easy to predict. First, we take a Bayesian perspective and propose the *Decompositional Disentangled Predictive Auto-Encoder* (DDPAE) as our formulation in Section 3.1. Next, we discuss our deep parameterization of each of the components in DDPAE in Section 3.2. Finally, we show how we learn the DDPAE by optimizing the evidence lower bound in Section 3.3.

### 3.1 Decompositional Disentangled Predictive Auto-Encoder

Formally, given an input video $x_{1:T}$ of length $T$, our goal is to predict future $K$ frames $\bar{x}_{1:K} = x_{(T+1):(T+K)}$. For simplicity, in this paper we denote any variable $\bar{z}_{1:K}$ to be the prediction sequence of $z$ from time step $T+1$ to $T+K$, i.e. $\bar{z}_{1:K} = z_{(T+1):(T+K)}$. We assume that each video frame $x_t$ is generated from a corresponding latent representation $z_t$. In this case, we can formulate the video frame prediction $p(\bar{x}_{1:K}|x_{1:T})$ as:

$$p(\bar{x}_{1:K}|x_{1:T}) = \iint p(\bar{x}_{1:K}|\bar{z}_{1:K})p(\bar{z}_{1:K}|z_{1:T})p(z_{1:T}|x_{1:T})\, d\bar{z}_{1:K}\, dz_{1:T}, \tag{1}$$

where $p(\bar{x}_{1:K}|\bar{z}_{1:K})$ is the frame decoder for generating frames based on latent representations, $p(\bar{z}_{1:K}|z_{1:T})$ is the prediction model that captures the dynamics of the latent representations, and $p(z_{1:T}|x_{1:T})$ is the temporal encoder that infers the latent representations given the input video $x_{1:T}$. From a Bayesian perspective, we model these three as probability distributions.

Our core insight is to decompose the video prediction problem in Eq. (1) into sub-problems that are easier to predict. In a simplified case, where each of the components can be predicted independently (*e.g.,* digits in Figure 1), we can use the following decomposition:

$$\bar{x}_{1:K} = \sum_{i=1}^{N} \bar{x}_{1:K}^i, \quad x_{1:T} = \sum_{i=1}^{N} x_{1:T}^i, \tag{2}$$

$$p(\bar{x}_{1:K}^i|x_{1:T}^i) = \iint p(\bar{x}_{1:K}^i|\bar{z}_{1:K}^i)p(\bar{z}_{1:K}^i|z_{1:T}^i)p(z_{1:T}^i|x_{1:T}^i)\, d\bar{z}_{1:K}^i\, dz_{1:T}^i, \tag{3}$$

where we decompose the input $x_{1:T}$ into $\{x_{1:T}^i\}$ and independently predict the future frames $\{\bar{x}_{1:K}^i\}$, which will be combined as the final prediction $\bar{x}_{1:K}$. We will use this independence assumption for the sake of explanation, but we will show later how this can easily be extended to the case where the components are interdependent, which is crucial for capturing interactions between components.

The key technical challenge is thus: How do we learn the decomposition? How do we enforce that each component is actually easier to predict? One can imagine a trivial decomposition, where

$x_{1:T}^1 = x_{1:T}$ and $x_{1:T}^i = 0$ for $i > 1$. This does not simplify the prediction at all, but only keeps the same complexity at a single component. We address this challenge by enforcing the latent representations of each component ($\bar{z}_{1:K}^i$ and $z_{1:T}^i$) to have low-dimensional temporal dynamics. In other words, the temporal signal to be predicted in each component should be low-dimensional. More specifically, we achieve this by leveraging the disentangled representation [6]: a latent representation $z_t^i$ is disentangled to the concatenation of (i) a time-invariant *content* vector $z_{t,C}^i$, and (ii) a time-dependent (low-dimensional) *pose* vector $z_{t,P}^i$. The content vector captures the information that is shared across all frames of the component. For example, in the first component of Figure 1, the content vector models the appearance of the digit "9". Formally, we assume the content vector is the same for all frames in both the input and the prediction: $z_{t,C}^i = \bar{z}_{t,C}^i = z_C^i$. On the other hand, the pose vector $z_{t,P}^i$ is low-dimensional, which captures the location of the digit in Figure 1.

This allows us to disentangle the prediction of decomposed latent representations as follows:

$$p(\bar{z}_{1:K}^i|z_{1:T}^i) = p(\bar{z}_{1:K,P}^i|z_{1:T,P}^i), \quad \bar{z}_t^i = [z_C^i, \bar{z}_{t,P}^i], \quad z_t^i = [z_C^i, z_{t,P}^i], \tag{4}$$

where the prediction $p(\bar{z}_{1:K}^i|z_{1:T}^i)$ is reduced to just predicting the low-dimensional pose vectors $p(\bar{z}_{1:K,P}^i|z_{1:T,P}^i)$. This is possible since we share the content vector between the input and the prediction. This disentangled representation allows the prediction of each component to focus on the low-dimensional varying pose vectors, and significantly simplifies the prediction task.

Eq. (2)-(4) thus define the proposed Decompositional Disentangled Predictive Auto-Encoder (DDPAE). Note that both the decomposition and the disentanglement are learned automatically without explicit supervision. Our formulation encourages the model to decompose the video into components with low-dimensional temporal dynamics in the disentangled representation. By training this structural generative model of future frames, the hope is to learn to produce good decomposition and disentangled representations of the video that reduce the complexity of frame prediction.

## 3.2 Model Implementation

We have formulated how we decompose the video prediction problem into sub-problems of disentangled representations that are easier to predict in our DDPAE framework. In this section, we discuss our implementation of each of the component of our model in Eq. (2)-(4), starting from the generation $p(\bar{x}_{1:K}^i|\bar{z}_{1:K}^i)$, inference $p(z_{1:T}^i|x_{1:T}^i)$, and finally prediction $p(\bar{z}_{1:K,P}^i|z_{1:T,P}^i)$.

**Frame Generation Model.** In Eq. (3), $p(\bar{x}_{1:K}^i|\bar{z}_{1:K}^i)$ is frame generation model. We assume conditional independence between the frames: $p(\bar{x}_{1:K}^i|\bar{z}_{1:K}^i) = \prod_{j=1}^{K} p(\bar{x}_j^i|\bar{z}_j^i)$. This model is used for both input reconstruction $p(x_t^i|z_t^i)$ and prediction $p(\bar{x}_t^i|\bar{z}_t^i)$. Our frame generation model is flexible and can vary based on the domain. For 2D scenes, we follow work in scene understanding [7] and use an attention-based generative model. Note that our latent representation is disentangled: $\bar{z}_t^i = [\bar{z}_C^i, \bar{z}_{t,P}^i]$, where $\bar{z}_C^i = z_C^i$ is the fixed content vector (*e.g.,* the latent representation of the digit), and $\bar{z}_{t,P}^i$ is the pose vector (*e.g.,* the location and scale of the digit). As shown in Figure 2(c), we generate the image $\bar{x}_t^i$ as follows: First, the content vector is decoded to a rectified image $\bar{y}_t^i$ using deconvolution layers. Next, the pose vector is used to parameterize an inverse spatial transformer $\mathcal{T}_z^{-1}$ [14] to warp $\bar{y}_t^i$ to the generated frame $\bar{x}_t^i$. The pose vector in this example is a 3-dimensional continuous variable, which significantly simplifies the prediction problem compared to predicting the full frame.

**Inference.** In Eq. (3), our prediction requires the *inference* of the latent representations, $p(z_{1:T}^i|x_{1:T}^i)$. Given our generation model $p(x_t^i|z_t^i)$, the true posterior distribution is intractable. Thus, the standard practice is to employ a variational approximation $q(z_{1:T}^i|x_{1:T}^i)$ to the true posterior [17]. Since our latent representations are decomposed and disentangled, we explain our model $q$ in the following two sections: Video Decomposition and Disentangled Representation.

**Video Decomposition.** The next question is: How do we get the decomposed $x_{1:T}^i$ from $x_{1:T}$? Eq. (3) assumes that the decomposition is given. Our key observation is that even if we decompose the input $x_{1:T}$ to $\{x_{1:T}^i\}$ in a separate step, the decomposed video would only be used to infer its respective latent representation through variational approximation. In this case, we can combine the video decomposition with the variational approximation as $q(z_{1:T}^i|x_{1:T})$, which directly infers the latent representations of each component. We implement $q(z_{1:T}^i|x_{1:T})$ using an RNN with 2-dimensional recurrence, where one recurrence is for the temporal modeling $(1:T)$ and the other is used to capture

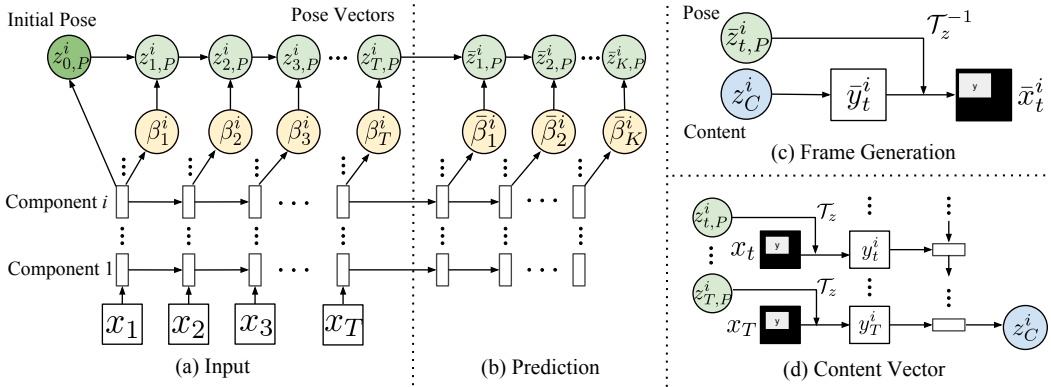

Figure 2: Overview of our model implementation. (a) We use 2D recurrence to implement $q(z^i_{1:T}|x_{1:T})$ to model both the temporal and dependency between components. (b) The prediction RNN is used only to predict the pose vector. (c) Our frame generation model generates different image with the same content using inverse spatial transformer. (d) A single content vector $z^i_C$ is obtained for each component from input $x_{1:T}$ and pose vectors $z^i_{1:T}$.

the dependencies between components. For instance, in the video in Figure 1, the component of digit "6" needs to know that "9" is already modeled by the first component. Figure 2(a) shows our 2-dimensional recurrence (our input RNN) in both the time steps and the components.

**Disentangled Representation.** While the 2D recurrence model can directly infer the latent representations, it is not guaranteed to output disentangled representation. We thus design a structural inference model to disentangle the representation. In contrast to frame generation, where the goal is to generate different frames conditioning on the same content vector, the goal here in inference is to *revert* the process and obtain a single shared content vector $z^i_C$ for different frames, and hence force the variations between frames to be encoded in the pose vectors $z^i_{1:T,P}$. Thus, we apply the inverse of the structural model in our generation process (see Figure 2(d)). For 2D scenes, this means applying the spatial transformer parameterized by $z^i_{t,P}$ to extract the rectified image $y^i_t$ from the frame $x_t$. We then use a CNN to encode each $y^i_t$ into a latent representation. Instead of training with similarity regularization [6], we use another RNN on top of the raw output as pooling to obtain a single content vector $z^i_C$ for each component. Figure 2(d) shows the process of inferring $z^i_C$ from $z^i_{1:T,P}$ and $x_{1:T}$. Since the same $z^i_C$ is used for each time step in prediction, this forces the decomposition of our model to separate the components with different motions to get good prediction of the sequence.

**Pose Prediction.** The final component is the pose vector prediction $p(\bar{z}^i_{1:K,P}|z^i_{1:T,P})$. Since $z^i_C$ is fixed in prediction, we only need to predict the pose vectors. Inspired by [16], instead of directly inferring $z^i_{t,P}$, we introduce a set of transition variables $\beta^i_t$ to reparametrize the pose vectors. Given $z^i_{t-1,P}$ and $\beta^i_t$, the transition to $z^i_{t,P}$ is deterministic with linear combination: $z^i_{t,P} = f(z^i_{t-1,P}, \beta^i_t)$. This allows us to use a meaningful prior for $\beta_t$. Therefore, as shown in Figure 2(a) and (b), given an input sequence $x_{1:T}$, for each component our model infers an initial pose vector $z^i_{0,P}$ and the transition variables $\beta^i_{1:T}$, from which we can iteratively obtain $z^i_{t,P}$ at each time step. We use a seq2seq [4, 32] based model to predict $\bar{\beta}^i_{1:K}$ (Figure 2(b)). With this RNN-based model, the dependencies between poses of components can be captured by passing the hidden states across components. This allows the model to learn and predict interactions between components, such as collisions between objects.

### 3.3 Learning

Our DDPAE framework is based on VAEs [17], and thus we can use the same variational techniques to optimize our model. For VAE, the assumption is that each data point $x$ is generated from a latent random variable $z$ with $p_\theta(x|z)$, where $z$ is sampled from a prior $p_\theta(z)$. In our case, the output video $\bar{x}_{1:K}$ is generated from the latent representations $\bar{z}^{1:N}_{1:K}$ of $N$ components, where $\bar{z}^i_t$ is the disentangled representation $[z^i_C, \bar{z}^i_{t,P}]$ (Eq. (4)) of the $i$th component, and $\bar{z}^i_{1:K,P}$ is parameterized by the initial pose $z^i_{0,P}$ and the transition variables $\beta^i_{1:(T+K)}$. Therefore, in our model, we treat $z^{1:N}_{0,P}$, $\beta^{1:N}_{1:(T+K)}$,

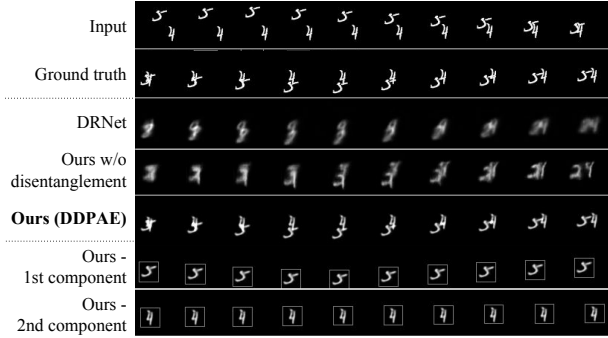

| | Input |
|---|---|
| | Ground truth |
| | DRNet |
| | Ours w/o disentanglement |
| | **Ours (DDPAE)** |
| | Ours - 1st component |
| | Ours - 2nd component |

Figure 3: DDPAE separates the two digits and obtains good results even when the digits overlap. The bounding boxes of the two components are drawn manually.

Table 1: Results on Moving MNIST (Bold for the best and underline for the second best). Our results significantly outperforms the baselines.

| Model | BCE | MSE |
|---|---|---|
| Shi et al. [43] | 367.2 | - |
| Srivastava et al. [31] | 341.2 | - |
| Brabandere et al. [5] | 285.2 | - |
| Patraucean et al. [25] | 262.6 | - |
| Ghosh et al. [10] | 241.8 | 167.9 |
| Kalchbrenner et al. [15] | **87.6** | - |
| MCNet [37] | 1308.2 | 173.2 |
| DRNet [6] | 862.7 | 163.9 |
| Ours w/o Decomposition | 325.5 | 77.6 |
| Ours w/o Disentanglement | 296.1 | 65.6 |
| Ours (DDPAE) | 223.0 | **38.9** |

and $z_C^{1:N}$ as the underlying random latent variables that generate data $\bar{x}_{1:K}$. We denote $\bar{z}$ as the combined set of random variables in our model. $\bar{z}$ is inferred from the input frames, $\bar{z} \sim q_\phi(\bar{z}|x_{1:T})$, where $q_\phi$ is our inference model explained in Section 3.2, parameterized by $\phi$. The output frames $\bar{x}_{1:K}$ are generated by $\bar{x}_{1:K} \sim p_\theta(\bar{x}_{1:K}|\bar{z})$, where $p_\theta$ is our frame generation model parameterized by $\theta$. Moreover, we assume that the prior distribution to be $p(\bar{z}) = \mathcal{N}(\mu, \text{diag}(\sigma^2))$. We jointly optimize $\theta$ and $\phi$ by maximizing the evidence lower bound (ELBO):

$$\log p_\theta(\bar{x}_{1:K}) \geq \mathbb{E}_q[\log p_\theta(\bar{x}_{1:K}, \bar{z}) - \log q_\phi(\bar{z}|x_{1:T})] = \mathbb{E}_q[\log p_\theta(\bar{x}_{1:K}|\bar{z}) - \text{KL}(q_\phi(\bar{z}|x_{1:T})||p(\bar{z})) \quad (5)$$

The first term corresponds to the prediction error, and the second term serves as regularization of the latent variables $\bar{z}$. With the reparametrization trick, the entire model is differentiable, and the parameters $\theta$ and $\phi$ can be jointly optimized by standard backpropagation technique.

# 4 Experiments

Our goal is to predict a sequence of future frames given a sequence of input frames. The key contribution of our DDPAE is to both decompose and disentangle the video representation to simplify the challenging frame prediction task. First, we evaluate the importance of both the decomposition and disentanglement of the video representation for frame prediction on the widely used Moving MNIST dataset [31]. Next, we evaluate how DDPAE can be applied to videos involving more complex interactions between components on the Bouncing Balls dataset [3, 36]. Finally, we evaluate how DDPAE can generalize and adapt to the cases where the optimal number of components is not known a priori, which is important for applying DDPAE to new domains of videos.

Code for DDPAE and the experiments are available at `https://github.com/jthsieh/DDPAE-video-prediction`.

## 4.1 Evaluating Decompositional Disentangled Video Representation

The key element of DDPAE is learning the decompositional-disentangled representations. We evaluate the importance of both decomposition and disentanglement using the Moving MNIST dataset. Since the digits in the videos follow independent low-dimensional trajectories, our framework significantly simplifies the prediction task from the original high-dimensional pixel prediction. We show that DDPAE is able to learn the decomposition and disentanglement automatically without explicit supervision, which plays an important role in the accurate prediction of DDPAE.

We compare two state-of-the-art video prediction methods without decomposition as baselines: MCNet [37] and DRNet [6]. Both models perform video prediction using disentangled representations, similar to our model with only one component. We use the code provided by the authors of the two papers. For reference, we also list the results of existing work on Moving MNIST, where they use more complicated models such as convolutional LSTM or PixelCNN decoders [15, 25, 43].

**Dataset**. Moving MNIST is a synthetic dataset consisting of two digits moving independently in a $64 \times 64$ frame. It has been used in many previous works [6, 12, 15, 24, 31]. For training, each

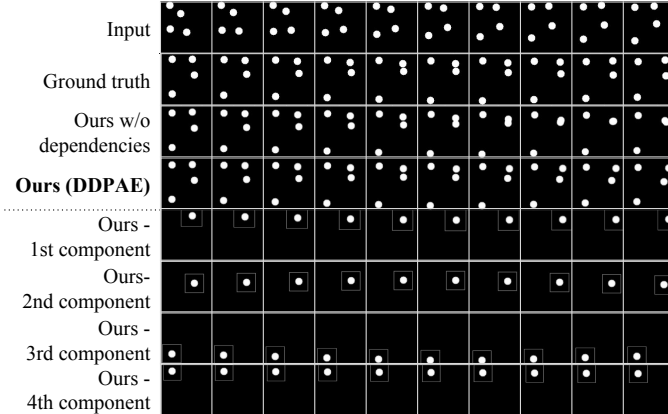

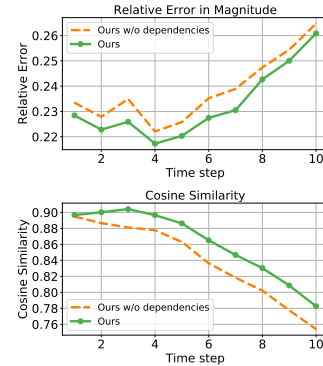

Figure 5: Accuracy of velocity with time. *Top*: Relative error in magnitude. *Bottom*: Cosine similarity.

Figure 4: Our model prediction on Bouncing Balls. Note that our model correctly predicts the collision between the two balls in the upper right corner, whereas the baseline model does not.

sequence is generated on-the-fly by sampling MNIST digits and generating trajectories with randomly sampled velocity and angle. The test set is a fixed dataset downloaded from [31] consisting of 10,000 sequences of 20 frames, with 10 as input and 10 to predict.

**Evaluation Metric.** We follow [31] and use the binary cross-entropy (BCE) as the evaluation metric. We also report the mean squared error (MSE) as an additional metric from [10].

**Results**. Table 1 shows the quantitative results. DDPAE significantly outperforms the baselines without decomposition (MCNet, DRNet) or without disentanglement. For MCNet and DRNet, the latent representations need to contain complicated information of the digits' combined content and motion, and moreover, the decoder has a much harder task of generating two digits at the same time. In fact, [6] specifically stated that DRNet is unable to get good results when the two digits have the same color. In addition, our baseline without disentanglement produces blurry results due to the difficulty of predicting representations.

Our model, on the other hand, greatly simplifies the inference of the latent variables and the decoder by both decomposition and disentanglement, resulting in better prediction. This is also shown in the qualitative results in Figure 3, where DDPAE successfully separates the two digits into two components and only needs to predict the low-dimensional pose vectors. Note that DDPAE can also handle occlusion. Compared to existing works, DDPAE achieves the best result except BCE compared to VPN [15], which can be the result of its more sophisticated image generation process using PixelCNN. The main contribution of DDPAE is in the decomposition and disentanglement, which is in principle applicable to other existing models like VPN.

It is worth noting that the ordering of the components is learned automatically by the model. We obtain the final output by adding the components, which is a permutation-invariant operation. The model can learn to generate components in any order, as long as the final frames are correct. This phenomenon is also observed in many fields, including tracking and object detection.

## 4.2  Evaluating Interdependent Components

Previously in Eq. (3), we assume the components to be independent, *i.e.,* the pose of each component is separately predicted without information of other components. The independence assumption is not true in most scenarios, as components in a video may interact with each other. Therefore, it is important for us to generalize to interdependent components. In Section 3.2, we explain how our model adds dependencies between components in the prediction RNN. We now evaluate the importance of it in more complex videos. We evaluate the interdependency on the Bouncing Balls dataset [3]. Bouncing Balls is ideal for evaluating this because (i) it is widely used for methods with access to physical states [2, 3, 9] and (ii) it involves physical interactions between components. One contribution of our DDPAE framework is the ability to achieve complex physics system predictions *directly from the pixels*, without any physics information and assumptions.

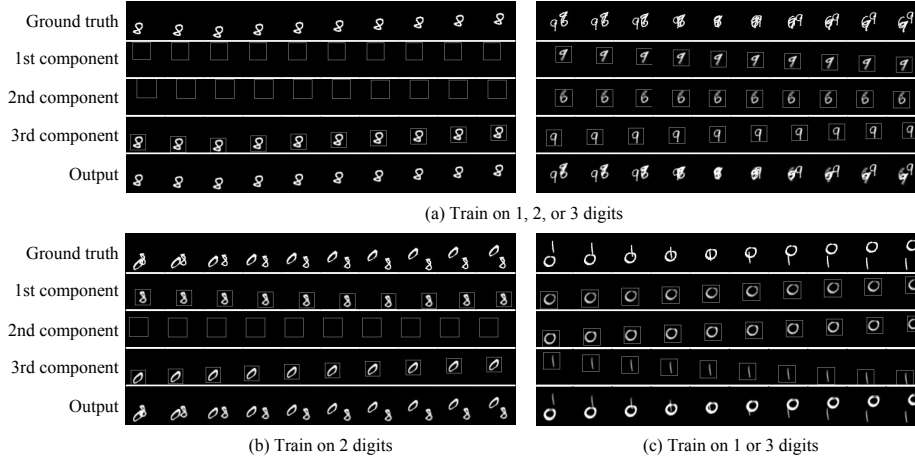

(a) Train on 1, 2, or 3 digits

(b) Train on 2 digits  (c) Train on 1 or 3 digits

Figure 6: Results of DDPAE trained on variable number of digits. Only the predicted frames are shown. Our model is able to correctly handle redundant components.

**Dataset**. We simulate sequences of 4 balls bouncing in an image with the physics engine code used in [3]. The balls are allowed to bounce off walls and collide with each other. Following the prediction task setting in [3], the balls have the same mass and the maximum velocity is 60 pixels/second (roughly 6 pixels/frame). The size of the original videos are 800 pixels, so we re-scale the videos to $128 \times 128$. We generated a fixed training set of 50,000 sequences and a test set of 2,000 sequences.

**Evaluation Metric.** The primary goal of this experiment is to evaluate the importance of modeling the dependencies between components. Therefore, following [3], we evaluate the predicted velocities of the balls. Since our model outputs the spatial transformer of each component at every time step, we can calculate the position $p_t^i$ of the attention region directly and thus the translation between frames. We normalize the positions to be $[0, 1]$, and define the velocity to be $v_t^i = p_{t+1}^i - p_{t-1}^i$. At every time step, we calculate the relative error in magnitude and the cosine similarity between the predicted and ground truth velocities, which corresponds to the speed and direction respectively. The final results are averaged over all instances in the test set. Note that the correspondence between components and balls is not known, so we first match each component to a ball by minimum distance.

**Results**. Figure 4 shows results of our model on Bouncing Balls. Each component captures a single ball correctly. Note that during prediction, a collision occurs between the two balls in the upper right corner in the ground truth video. Our model successfully predicts the colliding balls to bounce off of each other instead of overlapping each other. On the other hand, our baseline model predicts the balls' motion independently and fails to identify the collision, and thus the two balls overlap each other in the predicted video. This shows that DDPAE is able to capture the important dependencies between components when predicting the pose vectors. It is worth noting that predicting the trajectory after collision is a fundamentally challenging problem for our model since it highly depends on the collision surface of the balls, which is very hard to predict accurately. Figure 5 shows the relative error in magnitude and cosine similarity between the predicted and ground truth velocities, at each time step during prediction. The accuracy of the predicted velocities decreases with time as expected. We compare our model against the baseline model without interdependent components. Figure 5 shows that our model outperforms the baseline for both metrics. The dependency allows our model to capture the interactions between balls, and hence generates more accurate predictions.

## 4.3 Evaluating Generalization to Unknown Number of Components

In the previous experiments, the number of objects in the video is known and fixed, and thus we set the number of components in DDPAE to be the same. However, videos may contain an unknown and variable number of objects. We evaluate the robustness of our model in these scenarios with the Moving MNIST dataset. We set the number of components to be 3 for all experiments, and the number of digits to be a subset of $\{1, 2, 3\}$. Similar to previous experiments, we generate the training sequences on-the-fly and evaluate on a fixed test set.

Figure 6 (a) shows results of our model trained on 1 to 3 digits. The two test sequences have 1 and 3 digits respectively. For sequences with 1 digit, our model learns to set two redundant components to empty, while for sequences with 3 digits, it correctly separates the 3 digits into 3 components. We observe similar results when we train our model with 2 digits. Figure 6 (b) shows that our model learns to set the extra component to be empty.

Next, we train our model with sequences containing 1 *or* 3 digits, but test with sequences of 2 digits. In this case, the number of digits is unseen during training. Figure 6 (c) shows that our model is able to produce correct results as well. Interestingly, two of the components generate the exact same outputs. This is reasonable since we do not set any constraints between components.

## 5 Conclusion

We presented Decompositional Disentangled Predictive Auto-Encoder (DDPAE), a video prediction framework that explicitly decomposes and disentangles the video representation and reduces the complexity of future frame prediction. We show that, with an appropriately specified structural model, DDPAE is able to learn both the video decomposition and disentanglement that are effective for video prediction without any explicit supervision on these latent variables. This leads to strong quantitative and qualitative improvements on the Moving MNIST dataset. We further show that DDPAE is able to achieve reliable prediction *directly from the pixel* on the Bouncing Balls dataset involving complex object interaction, and recover physical properties without explicit modeling the physical states.

## Acknowledgements

This work was partially funded by Panasonic and Oppo. We thank our anonymous reviewers, John Emmons, Kuan Fang, Michelle Guo, and Jingwei Ji for their helpful feedback and suggestions.

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
