[Supplementary Material]

# Learning to Decompose and Disentangle Representations for Video Prediction (Supplementary)

**Jun-Ting Hsieh**
Stanford University
junting@stanford.edu

**Bingbin Liu**
Stanford University
bingbin@stanford.edu

**De-An Huang**
Stanford University
dahuang@cs.stanford.edu

**Li Fei-Fei**
Stanford University
feifeili@cs.stanford.edu

**Juan Carlos Niebles**
Stanford University
jniebles@cs.stanford.edu

## A   Implementation Details

For our image encoder and decoder, we use the DCGAN architecture [2] as the image encoder and decoder in our model. The number of layers are set based on the input or output image size, 5 layers for $64 \times 64$ images and 6 layers for $128 \times 128$. All recurrent neural networks are LSTMs with hidden size 64. The dimension of the content vector $z_C$ is 128, and the dimension of the pose vectors $z_{t,P}$ is 3, containing the parameters of a spatial transformer. We train our model for 200k iterations with the Adam optimizer [1] with initial learning rate 0.001, which is decayed to 0.0001 halfway through training. For all experiments, we optimize both the reconstruction and prediction losses during the first half of training, and optimize only the prediction loss in the second half, though we found that training with both losses throughout the entire training process produces similar results.

We assume our random latent variables, $z_{0,P}^i$, $\beta_t^i$, and $z_C^i$, to be Gaussian. Thus, our model outputs the mean and standard deviation for these variables. The prior distributions are $p(\beta_t^i) \sim \mathcal{N}(0, 0.1)$, and $p(z_C^i) \sim \mathcal{N}(0, 1)$. The prior for initial pose is $p(z_{0,P}^i) \sim \mathcal{N}([2, 0, 0], [0.2, 1, 1])$ for Moving MNIST, and $p(z_{0,P}^i) \sim \mathcal{N}([4, 0, 0], [0.2, 1, 1])$ for Bouncing Balls.

## B   Qualitative Results

In this section, we show more qualitative results on Moving MNIST (Figure 1) and Bouncing Balls (Figure 2). Figure 2 shows more examples where our model predicts the collision.

Below we present some failure cases for Bouncing Balls, where the balls fail to be separated. If the balls are too close together for all input frames, our model may produce blurry results. For collisions, since the trajectories after collision are highly sensitive to the collision surface, our model may identify the collision but produce incorrect trajectories.

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

Figure 1: Qualitative results on Moving MNIST.

Figure 2: Qualitative results on Bouncing Balls.

Figure 3: Bouncing Balls failure cases.