[Reviews · NeurIPS 2018]

Reviewer 1



The paper proposes a predictive model for video that decomposes the video into multiple components and probabilistically models the temporal dynamics of each component individually. Variational autoencoder is used to infer the approximate posterior of the latents. Temporal dynamics is modeled at the level of the latents (ie p(future frames latents | past frames latents)). Experimental results are nice, particularly on the bouncing balls data, showing that the model can capture the interactions between the components. There are some aspects that are unclear to me: (i) how is the final frame generated (sec 3.2)? by adding all components x^i_{1:k} for i=1,..n? (ii) what encourages the model to tease apart different components? How are z^i and x^i encouraged to be different for various 'i'? (iii) Fig 2: In the two dimensional RNN, one dimension runs over the components (vertical direction) which seems to impose arbitrary ordering over the components (ie 'i-1' comes before 'i'). Does it affect which components are captured in i < j? (iv) Line 182: what is the form of dynamical model 'f'? Minor: Line 218: "automatically" appearing twice =============== I have looked at the author response and am willing to maintain my original score.

Reviewer 2



This paper is well written and easy to follow. The authors tackle the problem of future video frame prediction (binary and simple MNIST and bouncing ball) and propose an improved variational autoencoder architecture for this task. The key insight is to model the time-invariant content, and the time-dependent content separately, and to force the time-dependent parameters (motion parameters), to be very low dimensional. The paper is complemented with experiments that demonstrate improvements of the previous state-of-the-art results. However the results are only tested on relatively low-dimensional and binary input frames of the MNIST and Bouncing ball dataset. Moreover, the authors needed to fix the number of possible objects from 1-3 in these experiments. It is unclear how the proposed architecture can scale beyond toy datasets. The authors note that "videos may contain an unknown and variable number of objects". There are many works in the ML community which also try to predict the future video frames such as: "Deep Predictive Coding Networks for Video Prediction and Unsupervised Learning", or "Deep multi-scale video prediction beyond mean square error ". As such it would be good to compare results on more realistic video.

Reviewer 3



The paper addresses the problem of predicting future frames in videos from previously seen frames. This task has been gaining a lot on popularity due to it's applications in different areas [8, 22], but also because it aims at modeling the underlying video representations in terms of motion and appearance. The paper proposes a new Decompositional Disentangled Predictive Auto-Encoder (DDPAE), a framework that combines structured probabilistic models and deep networks to automatically (i) decompose the high-dimensional video that we aim to predict into components, and (ii) disentangle each component to have low-dimensional temporal dynamics that are easier to predict. The main novelty is the automatic decomposition of the video into components that are easier to predict, as well as the disentanglement of each component into static appearance and 2D motion. Positive sides: 1. Well written and easy to read. The paper is well structured, the language maintains high standards and is easy to read. 2. Relevant and interesting task. The paper is tackling the problem of learning generative models of object dynamics and appearance in videos which is a challenging task with strong potential impact. 3. Novelty. Although the paper re-uses a lot of existing ideas and works, still the combination is novel and relevant to consider. 4. Experimental evaluation. The proposed model outperforms existing works with decomposition and disentanglement. Negative sides: 1. Clarity. The paper is really hard to follow and fully understand. Curently the paper spends little time in explaining the relevant details of this work: how is the video decomposed? What is the initial starting point when the video starts? How do you set the number of components in the video? How do you learn the content part of the disentangled representation? What is the parameterization of the pose part? How do you predict the betas? Why using a seq2seq model? Is the order between components arbitrary? If not (suggested by Fig 2) then how do you choose the order at training and test time? How do you ensure that the components correspond to objects (digits and balls)? How does the inference work? It is hard to grasp such a comlex part from only 3 lines of text. 2. Experiments. The experiments are done only in a very controlled setup with small and toy datasets. While that is fine as a debugging step to understand the model, ultimately the paper should tackle real-world datasets and showcase potential usability in the real world. Most of the maade assumptions (constant latent part) is fine on these toy datasets, but it brakes in real life. It is easy to define a priori the number of components in these toy datasets, because it is very constrained. In the real world the objects have much more complex pose dynamics, especially non-rigid objects. The datasets considered have black backgrounds only, which is a significant downside. You might hope that the proposed model in the real world might allocate a single (or several) components to the background? After reading the authors rebuttal, I remain with my decision to accept this paper. The rebuttal has been helpful to further understand the paper, which indicates that the paper is hard to thoroughly understand, however, the contribution is significant and this work will be definitely well accepted in the computer vision world.